# Development and Validation of the Adolescent Sexting Scale (A-SextS) with a Spanish Sample

**DOI:** 10.3390/ijerph17218042

**Published:** 2020-10-31

**Authors:** Cristian Molla Esparza, Pablo Nájera, Emelina López-González, Josep-Maria Losilla

**Affiliations:** 1Department of Research Methods and Educational Diagnosis, Faculty of Philosophy and Educational Sciences, University of Valencia, Avenida Blasco Ibáñez 30, 46010 Valencia, Spain; Emelina.Lopez@uv.es; 2Department of Social Psychology and Methodology, Faculty of Psychology, Autonomous University of Madrid, Cantoblanco Campus, 28049 Madrid, Spain; Pablo.Najera@uam.es; 3Department of Psychobiology and Methodology of Health Sciences, Faculty of Psychology, Autonomous University of Barcelona, Carrer Fortuna Edifici B, 08193 Bellaterra, Spain; JosepMaria.Losilla@uab.cat

**Keywords:** sexting, definition, measurement, validation, adolescents

## Abstract

“Sexting” is generally defined as the exchange of sexual media content via the internet. However, research on this topic has underscored the need to seek greater consensus when considering different conceptual elements that make up this definition. The aim of this study was to develop and validate an instrument for measuring sexting among adolescents, in order to cover a gap identified in the previous literature. The Adolescent Sexting Scale (A-SextS for short) was developed for validation on a sample of 579 Spanish secondary school pupils between the ages of 11 and 18. Evidence for face, content, concurrent, and criterion validity were assessed. A comprehensive set of 64 items, covering six defining characteristics of sexting (e.g., actions, recipient, media format, degree of sexual explicitness), was constructed after conducting an extensive literature review, two discussion groups, and a pilot study. Sexting prevalence rates measured by A-SextS were mostly concurrent with those found in previous studies. A-SextS subscales produced statistically significant positive associations with pornography consumption and physical sexual intercourse. The study shows that A-SextS can be an integrating instrument that facilitates a rigorous and comprehensive assessment of adolescent sexting experiences, as well as the formulation of an operationalized definition of the practice of sexting.

## 1. Introduction

“Sexting”, generally defined as the exchange of a message (hereinafter referred to as a “sext”) containing sexual content, produced by and commonly featuring the sender, via the communication means of the internet, has been shown to be prevalent among juveniles [1,2]. However, the paucity of theoretical explanations of consensus on this phenomenon has left an open debate about the motivations, opportunities, and risks of this practice [3]. Only a few exceptional studies have tried to relate sexting with existing psychological, social, and educative frameworks [4,5,6], though a growing literature supports the notion that sexting is a normative practice commonly used for sexual purposes [3,7,8,9]. From this perspective, sexting is understood as just another form of sexual expression in the context of contemporary sexual or romantic relations, which can, in fact, be carried out “safely” by young people when appropriate strategies are applied to reduce possible negative consequences [10]. The most common motivations for sexting cited by adolescents in the literature have been related to the initiation and/or the maintenance of incipient or established romantic relationships, whether in proximity or over long distances [3,11]. Such motivations comprised, for example, the intention to attract attention, to flirt, to develop sexual interest, or to initiate a real-life sexual experience [6,12,13,14]. Other less reported and understudied motivations related to social purposes, such as having fun, joking, and killing time, to identity construction, such as self-expression and body image acceptance, or to peer group influences, such as imitation or gaining acceptance [8]. Although a number of studies consider sexting as a normative sexual behavior among young people, they also acknowledge that it comes with certain risks [15,16]. A common risk is the intentional, non-consensual distribution of third-party sexual images, whose prevalence among youths has been shown to lie between 8.4 and 15.6% [2].

Adolescent sexting prevalence rates are extremely heterogeneous, and correlates are still inconclusive, especially concerning demographic variables. Nevertheless, meta-analysis has suggested a mean prevalence of sexting, in terms of sending and receiving sexts, of 14.8% (95% CI: 12.8%, 16.80%; *I*^2^ = 99%) and 27.4% (95% CI: 23.1%, 31.7%; *I*^2^ = 98.7%), respectively, with a high heterogeneity of results [2]. In an ongoing meta-analysis it has also been observed that such rates have been progressively increasing over the last ten years (e.g., 7%, 95% CI: 5%, 10% for sending sexts in studies collecting data in 2009, versus 16%, 95% CI: 13%, 20% in 2014, and 29%, 95% CI: 20%, 39% in 2018) [17]. The most supported findings concerning correlates have suggested that sexting is more prevalent with increasing age [2], and is significantly related to adolescent sexual behavior, such as having actual sexual intercourse, and to other online and offline sexual experiences, such as pornography consumption [18,19]. The disparate findings in sexting research may be due to differences in how the practice has been conceptualized and measured among the various studies [2,3,20,21]. Several critical review studies on sexting have identified up to six elements constituting its definition and have revealed substantial differences in its assessment, in the actions that it entails, in the willingness to partake in it, in the recipients and audiences, in the media content transmitted, in the libidinous character of the contents, and in the timeframe of the measure [3,21]. Barrense-Dias et al. [21] highlighted that some studies considered active experiences of sexting, such as sending, asking for, or posting sexts, while others also included passive experiences, such as being asked for, or having receiving sexts. Such actions were sometimes reported separately and sometimes combined in one item. Some studies have also distinguished “primary sexting”, when a person sends their own personal sexts to others, from “secondary sexting”, which implies the further dissemination of such material without the consent of the person referenced by the sext [22,23]. Although sexting is often thought of as a voluntary practice [24,25], most studies do not specify it as such [17], nor consider the indirect pressure to exchange sexts that adolescents may feel or receive [26]. There are also differences in methods of transmission, such as using a mobile phone, a computer, email, or an unspecified method [21]. The format of and terms used to define sexts can also differ between studies [21]. While some definitions have only considered text messages, others have additionally included audio-visual content (e.g., images or videos), without analysing them separately [17]. The majority of articles have characterized sexts using very general adjectives, such as “sexy”, “sexual”, and “provocative”, while only a minority have seen them adjectivized using more precise terms such as “nude” or “wearing only underwear” [21]. Only a few studies have assessed and reported sexting considering different addressees, such as partners, acquaintances, strangers, and so on [25,27]. This deficiency in the literature can be considered especially important when dealing with adolescents, since the risks they are exposed to may vary with different sexting recipients [21,28]. Gámez-Guadix et al. [25], for example, found that the relationship between sexting and online sexual victimization (OSV) was stronger when sexts were sent to a person met only online. The purpose behind exchanging sexts in such contexts and with such recipients is an aspect that has been considered in very few studies [29,30]. Another conceptual consideration is that the timeframe of measures used to assess sexting has varied considerably across studies, with some accounting for a month prior to surveying, and others referring to lifetime prevalences [31,32]. Lastly, the most notable methodological limitation of research on sexting is the absence of a consensus on its measurement, especially in adolescents [17].

### 1.1. Existing Validated Sexting Measures

Several recently validated instruments have focused on assessing sexting in adults, such as the “Escala de Conductas sobre Sexting” (ESC, only available in Spanish at this time) [33] and the Sexting Behavior Scale (SBS) [34]. However, to the best of our knowledge, to date, there has only been one instrument tested among adolescents, the Intimate Images Diffusion Scale (EDIMA) [35]. EDIMA was validated on a sample of Spanish adolescents (602 adolescents aged between 12 and 19) in order to estimate the frequency of sending and distributing suggestive or provocative images or videos via mobile phones. In particular, the scale refers to four actions (sending, receiving, requesting, and re-sending) and distinguishes between three possible agents (partners, friends, and acquaintances, and strangers). The reliability score of EDIMA was found to be 0.976.

Despite efforts to advance the assessment of sexting, most of the aforementioned conceptual and methodological issues have still not been resolved. The scales employed thus far suggest that sexting occurs via mobile phones or social network sites without covering other potential technologies or platforms. Nor do they cover other possible media formats, such as audio [31,36]. Sexts have thus far only been characterized with very general adjectives, and, therefore, have been subject to the interpretation of respondents. Lastly, voluntariness has not been expressly considered, making it impossible to distinguish between fine-grained degrees of voluntariness in sexting, such as intentional sexting, unwanted but consensual sexting, and coerced sexting [17].

### 1.2. The Purpose of the Present Study

The aim of this study was to develop and validate a new instrument for measuring sexting among adolescents that would cover the wide range of conceptual and methodological aspects mentioned above. In particular, the innovative nature of this instrument, which we have named the Adolescent Sexting Scale (A-SextS for short), lies in considering, unifying, and clarifying the variety of conceptual reference elements in the definition of sexting in terms of: (a) focusing on active sexting, covering a wide range of online behaviors, some of which have not been considered to date (e.g., posting, streaming); (b) distinguishing with the same motivational framework (i.e., with amorous or sexual purpose) up to three different types of addressee; (c) differentiating whether sexting occurs due to the participant’s own initiative or not; (d) considering as wide a range of media formats as possible (e.g., sexts, beyond text, and visual formats); (e) distinguishing three degrees of sexual explicitness in sexts; and (f) examining other relevant information such as whether the face of the participant appears in the images or videos sent. Furthermore, our study moves away from a conventional validation strategy towards a theory-driven approach, with a known factor structure, since sexting still cannot be considered a construct of its own embedded in a validated theoretical system, but rather simply a system of behaviors carried out via the internet. Therefore, at the current stage of research on sexting, a validation approach focused mainly on content validity, concurrent validity, and criterion validity is required. Thus, our study makes a three-fold contribution to advancing sexting research: further theoretical and empirical development, more accurate prevalence estimates, and a more complete characterization of the practice specifically among adolescents.

## 2. Method

### 2.1. Participants

Data for the study was obtained from a convenience sample composed of 579 adolescents (305 males and 274 females), aged 11–18 years (M = 13.9 years; SD = 1.3), from two secondary charter schools located respectively in a metropolitan and a rural area of the Autonomous Community of Valencia, in Spain. The sample included 161 (27.8%) seventh grade students, 162 (28%) eighth graders, 144 (24.9%) ninth graders, 94 (16.2%) tenth graders, and 18 (3.1%) basic vocational training students. The participants’ age range distribution is presented in Table 1.

### 2.2. Procedure

The administrations of the schools were contacted by email in order to arrange meetings and explain the study’s goals and ethical procedure. The school principal, together with the school board members, decided whether the school would participate. One of the schools decided to collaborate only in carrying out the pilot test of the scale, whereas the other two schools participated in the final data collection. Through the letter of consent, parents were also informed of the ethical procedure of the study, of the content of the questionnaire, and of their right to refuse the participation of their children, which occurred in only four cases. No agent required clarification or suggested modifying any of the questions. The data collection took place between 2nd March 2020 and 13th March 2020. The questionnaire was administered to the participating adolescents in their usual classrooms, during regular class hours, and took approximately 40 minutes. Participants received all the instructions via a video tutorial recorded by real professional speakers, and then recorded their own responses on paper questionnaires (see Appendix B for the original video tutorial administered, and Appendix A for the transcription of these instructions). The adolescents were informed that participation in this research project was entirely voluntary, and no negative consequences would result from them abandoning or not participating it in. Ultimately, no adolescent abandoned or refused to participate in the project. The present research was performed according to the Institutional Review Board (IRB) guidelines [37] and with the current Spanish laws on the Protection of Personal Data and guarantee of digital rights (LO 3/2018 of 5 December). Participants did not receive any compensation.

### 2.3. The A-SextS Development and Validation Process

According to the background and the purpose of the instrument, our study was carried out in two stages. In the first stage, content and face validity were addressed via three strategies: (a) conducting an extensive literature review; followed by (b) conducting adolescent discussion groups; and (c) conducting a pilot study. The second stage was then aimed at obtaining a set of concurrent and criterion validity evidences regarding the instrument.

#### 2.3.1. Stage 1: Content and Face Validity

Content and face validity were addressed via a review of measures applied in empirical studies on the prevalence of sexting with juvenile samples and published between 2009 and 2020. A total of 79 studies were included in our review (See Appendix C) [17]. This extensive review of sexting measures allowed this study’s authors to identify a wide range of conceptual reference elements used to constitute the operational definition of sexting. In particular, for the purpose of this study, the following data was recorded: (a) measure quality (i.e., whether the study reported evidence of validity and/or reliability in the study sample or in comparable samples); (b) the elements making up each definition of sexting (e.g., experiences, media formats); (c) whether the study identified specific addressees or recipients; (d) whether a goal or purpose for sexting was specified; (e) the number of primary items used to assess sexting; (f) whether a single or combined measure was used (i.e., two or more actions at a time); (g) response types (e.g., dichotomous, the Likert frequency scale); and (h) the number of response categories (e.g., four, five-point Likert scale). As a result, the research team created and revised an initial pool of domains and items that could be used in the discussion groups. The qualitative findings of this review have already been presented in the Introduction of this paper, and were used to generate a satisfactory conceptual framework for the practice of sexting. Quantitative findings will be reported in the following Results section.

After the extensive literature review, two discussion groups were conducted to examine content and semantic validity, characterizing some domains and clarifying certain wordings and terminologies that previous literature had defined as inconsistent and vague [23]. One discussion group consisted of 10 participants (6 females of ages 11–12), while the other consisted of 11 participants (8 females of ages 11–12). Both discussion groups were conducted and guided by the first author in order to collect new potentially relevant items and to adapt A-SextS according to the participants’ suggestions. The discussion group guide had been previously pilot-tested, and addressed the possession, use, and supervision of information and communication technologies (ICT) and social networks, the typology and characterization of social relations, aspects of sexuality and sexting, and the characterization of sexual multimedia content (see Appendix A for further information). Semi-structured open-ended questions and multiple-choice question activities were also proposed to the participants. During the discussions, written notes were taken to allow post-event review by the authors E.L.G. and J.-M.L. Each discussion lasted 60 min. The process resulted in several changes in wording of certain items and the deletion of others that were not deemed relevant or were considered uncommon. For example, certain adjectives used in some of the items specifying the explicitness of the media content were changed to improve comprehension of the items by adolescents (e.g., “sending audios of a sexual nature” was changed to “sending sexy audios”). The expression initially used to refer to sexts featuring someone else was also changed (e.g., “I have sent an image or video where other nudes appear” was changed to “I have sent a sexy image or video featuring someone else”). Certain terms were also adapted to incorporate adolescent jargon (e.g., “I have broadcast a video” was changed to “I have live-streamed video”). Items that were not deemed relevant or were considered uncommon were deleted (e.g., posting or live-streaming sexy audios, asking someone to do live broadcasts nude, in underwear, or dressed and in a sexy pose).

Finally, A-SextS’ updated list of 67 questionnaire items was pilot-tested on 96 secondary school pupils. After completing the pilot-questionnaire, the adolescents were asked about the readability and comprehension of the questionnaire, initiating a brief oral discussion between the researcher and the adolescents. Written notes were taken for decision-making purposes. The pilot test provided useful insights as to how improve the instructions, appearance, and format of the questionnaire. For example, the sociodemographic questions section was moved to the end of the questionnaire to avoid a possible fatigue effect at the time of filling it in and to prevent its answers from being conditioned by having provided such information previously. The items in the questionnaire were changed from a bulleted or numbered format (e.g., “I have sent a sexy text message to: (a) my boy/girlfriend”, and so on, followed by the frequency scale for each one) to an unnumbered format with a full text sentence. A reminder of the basic instructions in the top margin of the scale was also added to the final version. Ambiguous items were also discussed with the pupils and modified where deemed necessary. These pilot test participants were not included in the final sample. The final version of A-SextS was composed of 64 items.

#### 2.3.2. Stage 2: Concurrent and Criterion Validity

Concurrent validity, which is the extent to which the results of the scale agree with other, independent external results, was examined by comparing our A-SextS’ prevalences in this study’s sample with prevalence estimates reported in previous meta-analytic studies or similar individual empirical studies. For this purpose, a comprehensive literature review was performed (see Appendix A to consult the databases and search strategy used). After conducting the literature review, our prevalences were compared with the estimates of the review and meta-analytical studies on sexting prevalence carried out by Klettke et al. [20] and Madigan et al. [2]. They were also compared with the results of an ongoing meta-analysis, including studies up until February 2020, providing prevalence rates clustered by year of data collection and further exploring key elements making up the operational definition of sexting [17]. Among the available empirical literature on sexting prevalence, we found four comparable studies that included similar samples and distinguished between addresses in their instruments. The empirical studies selected were those of Burén and Lunde [38], Schloms-Madlener [39], Dolev-Cohen and Ricon [40], and Quesada et al. [41] To compare sexting experiences not defining a specific addressee (e.g., posting), we considered the empirical studies of Gregg et al. [29], Jonsson et al. [42], Kerstens and Stol [43], and Kopecky [44].

Criterion validity was supported by relations between A-SextS and different variables in the available literature. This was the case for age, sexual activity, and pornography consumption, which have consistently been found to positively correlate with sexting [2,18,19]. Nine different subscales of A-SextS were also defined according to sexting action and addressees: sending sexts to a boy/girlfriend (SF), sending sexts to someone known in person (SK), sending sexts to someone known only on internet (SI), posting or live-streaming pictographic content (PS), asking for sexts from a boy/girlfriend (AF), asking for sexts from someone known in person (AK), asking for sexts from someone only known on the internet (AI), receiving sexts (R), and refusing to send sexts (RS).

### 2.4. Measurement

#### 2.4.1. The Adolescent Sexting Scale (A-SextS)

“Sexting” is given to mean the exchange of sexy media content over the internet for an amorous or sexual purpose. This definition encompasses the existence of various different experiences (e.g., sending, receiving), addressees and audiences (e.g., partners, potential partners), media formats (e.g., texts messages, videos), degrees of explicitness (e.g., nude, semi-nude), degrees of willingness (e.g., whether or not sexting occurs due to one’s own initiative), and degrees of privacy (e.g., whether the participant shows their face or any other personally identifiable part of their body in the content). For the validation of our own Adolescent Sexting Scale (A-SextS), “sexting” was thus briefly defined to respondents, “as a term, given to mean the exchange of sexy text messages, audios, images, or videos over the internet with another person, and doing it with an amorous or sexual purpose”. A-SextS was focused on a representative range of active sexting experiences, but also included the reception of sexts as the main passive experience. A-SextS was composed of (a) three synchronous experiences (audio calls with 3 items; video calls with 9 items; and live-streaming with 3 items), (b) four asynchronous experiences (sending with 18 items; posting with 4 items; refusing a request to send with 3 items; and receiving with 3 items), and (c) one experience that could imply both (asking to be sent with 21 items). A-SextS distinguished between four media formats (text messages, images, videos, and audios), two possible protagonists of the media content (oneself or another person), and three possible addressees (boyfriend/girlfriend, someone I know in person, and someone I only know on the internet). A-SextS, in terms of pictographic content, also distinguished between three levels of sexual explicitness (naked, in one’s underwear, and dressed and in a sexy pose). The final version of A-SextS was composed of 64 items using a 5-point Likert-scale (0 = Never to 4 = More than once a day), inquiring about sexting experiences during the 30-day period prior to taking the questionnaire. Additionally, in the case of affirmative answers, (a) participants were invited to indicate, via a multiple-choice question, whether the action was done due to their own initiative or in response to a request, and (b) in reference to pictographic media content, whether they showed their face in the content or not. The final version of the questionnaire (in Spanish as administered to participants, but also made available in English) is provided in the Appendix D, while a slightly refined version is provided in Appendix E. A summary of the items, together with response percentages, is shown in Table 2. The modular structure of the questionnaire is then summarized graphically in Figure 1, and described in detail in Appendix A.

#### 2.4.2. Socio-Demographic Variables

Participants were asked to indicate their biological sex, age, sexual orientation, and parental cohabitation status.

#### 2.4.3. Dating Scenarios

Participants were asked to indicate whether they “like or have liked someone” and whether they “have or have had a boyfriend/girlfriend”, and, for both questions, the age of that person.

#### 2.4.4. Sexuality, Sexual Experiences, and Sexual Risk-Taking Behaviors

The questionnaire asked adolescents how they found information on sexuality, as well as whether they used mobile apps for hooking up with people, whether they watched pornography and how many times a week they did so, whether they had felt pleasure masturbating and, again, how many times a week, whether they had had sex with penetration, and whether any sexual relations had occurred without them wanting it. One additional item was used to assess how attractive they considered themselves to be. Finally, participants were asked about any perceived consequences of sexting.

### 2.5. Data Analysis

In order to assess concurrent validity, A-SextS’ single items, comprising all conceptual elements (e.g., addressee, media format, sexual explicitness), were considered the main comparison domains. When comparison was not possible due to divergences in conceptual elements constituting the sexting measurement, broader and less refined domains were considered according to the operational definition of the comparison study. Experiences and the addressees of sexting were considered as the two most basic defining elements for the comparisons. In certain studies, additional calculations were made to determine proportions and 95% confidence intervals (95% CI). Satisfactory concurrent validity was thus found to be supported by overlapping confidence intervals.

To assess criterion validity, first of all, the essential unidimensionality of each of the nine subscales was checked using different criteria: a parallel analysis with principal components, polychoric correlations, and the mean criterion [45], the number of eigenvalues higher than one, and a ratio between the first and second eigenvalue higher than 4 [46]. Item scores were summed to create an aggregated subscale, and confirmed as essentially unidimensional [46]. Furthermore, average communalities, Cronbach’s Alpha [47], and McDonald’s Omega [48] were also evaluated for each subscale. Secondly, Kendall’s Tau coefficient [49], between each subscale and each criterion variable (i.e., age, pornography consumption, and physical sexual activity), was computed. Both the magnitude and statistical significance of the correlations were then assessed. Regression analyses were also conducted to examine the effect of pornography consumption and physical sexual activity on each subscale. Both variables were dichotomous (0 = have never consumed pornography/had sexual relations; 1 = have consumed pornography/had sexual relations at least once). Given the positive skewed distribution of the subscale scores, three different types of regressions were conducted: binary logistic regression, Poisson regression, and negative binomial regression. For the binary logistic regression, the subscale scores were dichotomized in either 0 (e.g., have never sent a sext to one’s boyfriend/girlfriend) or 1 (e.g., have sent at least one sext to one’s boyfriend/girlfriend). For the Poisson regression, the item scores were recategorized to approximate a frequency metric. The recategorized frequency scores were 0, 2, 8, 30, and 60 for each of the 5 scale categories, respectively. For instance, a score of 8 corresponded to category 2 (i.e., between 1 and 3 times a week), averaging out at 8 times a month. For the Poisson regression, the subscales reflect an aggregation of these recategorized items. For the negative binomial regression, the subscale scores reflected the sum of the item direct scores (from 0 to 4). For each of the three regression types, the effect of pornography consumption and physical sexual activity on each of the nine different subscales was assessed, and the effect of both variables was corrected by gender and age. As an example, the binary logistic regression model to examine the effect of pornography consumption (corrected by gender and age) on the (dichotomized) SF subscale was calculated as: $logit\left(SF=1\right)=\beta_{0}+\beta_{1}Pornography+\beta_{2}Gender+\beta_{3}Age$. A total of 54 regression analyses (3 types × 9 subscales × 2 criterion variables) were conducted. The exponentiated regression coefficients (ExpB), as well as their 95% CI and statistical significance, were also calculated.

All analyses were performed using the statistical software R [50] and the packages “sirt” version 3.9–4 [51], “psych” version 2.0.8 [52], and “MASS” version 7.3–53 [53]. Additional analyses were performed using IBM SPSS Statistics 25 [54].

## 3. Results

### 3.1. Evidence of Content and Face Validity

#### 3.1.1. Critical Systematic Review of Sexting Measures

Quantitative results of the 79 studies relating to sexting prevalence included in our review (see section “Stage 1” above) showed that mobile phone usage is the most referenced communication channel when asking about sexting (*n* = 28). Many studies ask about sending sexts (*n* = 69). Images (*n* = 74) and videos (*n* = 39) were the most considered media format and “sexual” (*n* = 30) was the most common adjective used to characterize sexts. Most studies did not make explicit the timeframe of the measure (*n* = 45). Only a minority of studies (*n* = 16) considered the addressee (for those sending) or the sender (for those receiving) and only 2 studies defined the purpose of sexting among primary items. The response formats most used to assess sexting were the Likert scale (*n* = 28) and dichotomic responses (*n* = 24). Among studies considering the act of sending, most evaluated it via a mono-item (*n* = 34). Lastly, only *n* = 21 reported any reliability index or evidence of the validity of measures applied. See Appendix A for more details.

#### 3.1.2. Discussion Group Results

Discussion groups characterized up to three different types of social relationships: (a) friendships with daily contact, trust, and esteem, (b) dating relationships, including the characteristics of friendship in addition to attraction and exclusivity, and (c) relationships with people they know only on the internet, with whom they have less contact and about whom they do not know very much. The difference between a person you only know on the internet and a stranger was considered to be that the latter you do not talk regularly with, you have never talked to, or have never intended to talk to, and you do not have any information about. The discussion groups agreed that “private parts” referred to genitals and the backside as intimate parts of both boys’ and girls’ bodies, with the addition of breasts as intimate parts of girls’ bodies. The discussion groups considered that all potential actions were covered in evaluating A-SextS. On the basis of their answers, adjectives that best described (a) nude pictographic contents were “naked” (*n* = 7) and “showing penis/vagina, breasts, and/or backside” (*n* = 4), (b) semi-nude contents were “covered by underwear” (*n* = 6) and “almost naked” (*n* = 5), and (c) neither nude nor semi-nude were “dressed and in a sexy pose” (*n* = 13) and “seductive” (*n* = 3). See Appendix A for more details.

### 3.2. Evidence of Concurrent Validity

Compared to review studies on sexting prevalence, the overall prevalence of sending sexts reported in this study (95% CI: 49.9%, 58%) was higher than that reported by Klettke et al. [20] (95% CI: 2%, 19%) and Madigan et al. [2] (95% CI: 13%, 17%), and slightly higher than the stratified estimate in studies collecting data in 2018 (95% CI: 22%, 46%) considered in our ongoing meta-analysis [17]. Our reception rate of sexts (95% CI: 43%, 71%) was also higher than that of Klettke et al. [20] (95% CI: 11.7%, 19.6%) and Madigan et al. [2] (95% CI: 23.1%, 31.7%), but was consistent with the stratified estimate in studies collecting data in 2018 (95% CI: 22%, 46%) considered in our ongoing meta-analysis [17].

Results of empirical studies with samples of Spanish adolescents, such as the estimates of Quesada et al. [41], for the voluntary sending of sexual images or videos to a partner (95% CI: 15%, 24.1%), to a friend/acquaintance (95% CI: 8%, 15.3%) and to someone only known on the internet (95% CI: 4.9%, 11%), coincide with those obtained in our study when pooling degrees of explicitness (95% CI: 20.3%, 28.6%; 14.3%, 20.4%; and 2.4%, 5.5%, respectively). The estimate for the voluntary sending of text messages with sexual content to someone only known online (95% CI: 3.2%, 8.4%) also concurs with our estimation (95% CI: 7.2%, 12%). However, when the addressee is an established partner (95% CI: 18.4%, 28.1%) or a friend/acquaintance (95% CI: 11.5%, 19.8%), our estimates are slightly higher (95% CI: 31.4%, 40.6% and 22.8%, 30%, respectively).

The prevalences reported by Schloms-Madlener [39] regarding the sending of suggestive text messages to an established partner (95% CI: 37%, 56.2%) and to someone known only online (95% CI: 11.1%, 17.5%) are consistent with those in our study (95% CI: 38.6%, 48.1%; and 12.2%, 18%). Schloms-Madlener’s [39] estimate of sending nude or semi-nude images to someone known only online (95% CI: 3.4%, 7.5%) also coincides with that reported in our study (95% CI: 6.4%, 10.9%). Our estimate is slightly higher when the addressee is an established partner (95% CI: 13.5%, 29.2%), but the intervals are close (95% CI: 32.4%, 41.8%). This is similarly the case in comparison with Dolev-Cohen and Ricon’s [40] estimate regarding the same addressee, experience, and content (95% CI: 12.3%, 18.9%). The estimated rate of asking another person to send a nude photo in the Dolev-Cohen and Ricon [40] study (95% CI: 9.5%, 15.6%) is also consistent with our pooled estimate (95% CI: 9.4%, 14.6%), despite our measure also including video content. Our estimate does not coincide with that obtained by Dolev-Cohen and Ricon [40] when the person requested to send a nude photo is an established partner (95% CI: 3.7%, 7.9%), but is relatively close (95% CI: 9.2%, 15.5%).

In the study by Burén and Lunde [38], the prevalence of sending sexual images or videos (including webcam videos) to romantic partners (95% CI: 13.4%, 16.8%) was lower compared to the rate obtained in our sample (95% CI: 26.4%, 33.9%), whereas, when the addressee is an online friend/acquaintance (95% CI: 4.3%, 6.5%), our prevalence coincides (95% CI: 6.4%, 10.9%).

Kopecky’s [44] estimate of adolescents posting sexy photos or videos partially or completely naked on the internet (95% CI: 6.3%, 9.2%) coincides with our estimate (95% CI: 9.2%, 14.5%). The prevalence of posting sexual images on the internet reported by Gregg et al. [29] (95% CI: 11.2%, 19%) is also consistent with the one obtained in our study, also considering video content (95% CI: 9.2%, 14.5%). Jonsson et al. [42] found a prevalence of adolescents posting pictures or films partially undressed of 9.8% (95% CI: 8.9%, 10.9%), which is also consistent with our estimate (95% CI: 8.6%, 13.7%).

Kerstens and Stol’s [43] estimate of adolescents’ exposure of breasts and/or genitals via webcam (95% CI: 1.2%, 2%) concurs with our prevalence of nude video live-streaming (95% CI: 0.6%, 2.5%). The prevalence of sexual exposure or “flashing” via webcam or mobile phone (95% CI: 13.2%, 15.6%) reported by Jonsson et al. [42] is also consistent with that reported in our study (95% CI: 8.2%, 13.2%).

To the best of our knowledge, there have been no studies measuring the outstanding sexting experience in adolescents, that is, the sending of sexy audio recordings. Thus, estimates obtained in this study could not be compared to the previous literature.

### 3.3. Evidence of Criterion Validity

Table 3 shows essential unidimensionality results for the nine subscales. All the subscales obtained the essential unidimensionality criteria. Exceptionally, the subscale R (receiving sexts) obtained an eigenvalue ratio of 3.9, very close to the cut-off criteria set at 4. In addition, the subscales showed high average communalities ($h^{2}\geq 0.55$) and reliability (α≥0.62, ω≥0.81).

Table 4 shows Kendall’s Tau coefficient between each subscale and the three criterion variables. Most of the relations were positive and statistically significant, except for age with SI and AK, and pornography consumption with PS and AK. All nine subscales obtained positive and statistically significant associations with physical sexual activity.

Finally, Table 5 shows the exponentiated coefficients (ExpB) resulting from the regression analyses. Both pornography consumption and physical sexual activity obtained a positive and statistically significant effect on all nine subscales, regardless of the regression type and after correcting for gender and age. Among these results, especially strong relations were found between pornography consumption and asking for sexts to someone only known online (ExpB binary logistic: 4.1; ExpB Poisson: 12.1; ExpB negative binomial: 7.3), as well as between physical sexual activity and sending sexts to one’s boyfriend/girlfriend (ExpB binary logistic: 7.6; ExpB Poisson: 4.5; ExpB negative binomial: 4.1), asking for sexts from one’s boyfriend/girlfriend (ExpB binary logistic: 7.6; ExpB Poisson: 9.2; ExpB negative binomial: 9.3), and asking for sexts from someone known in person (ExpB binary logistic: 4.6; ExpB Poisson: 13.6; ExpB negative binomial: 7).

## 4. Discussion

The goal of this study was to meet the need to create a measure of sexting that integrates and clarifies a wider variety of conceptual elements constituting its definition. Furthermore, the study aimed to overcome conceptual and methodological shortcomings detected not only in previous empirical studies, but also in previous scales developed on this topic (e.g., the non-consideration of certain sexting experiences and media formats, the use of vague adjectives to describe sexts, the non-specification of certain elements, such as purpose and the temporal framework) [2,3,17,20,21,33,34,35]. To achieve the above goals, our Adolescent Sexting Scale (A-SextS for short) was based on multiple sources of information, including a thorough literature review, discussion groups, and a pilot study.

Regarding content and face validity, A-SextS has several notable strengths in comparison to previous measures deserving consideration. First, our instrument went a step further than others by focusing on active and primary sexting, and by considering actions that had not been taken into account to date, such as the live-streaming of content. Second, our scale also expands upon current work on sexting measures by including hitherto unconsidered media formats, beyond text message and visual formats, such as audio recordings. Third, this is one of the first measures that characterizes pictographic sexts precisely by objectively differentiating three degrees of sexual explicitness and considers relevant information such as whether the sender shows their own face in the content or not. Fourth, as several scholars have argued that the risks young people engaged in sexting are exposed to may differ according to the recipient of the content [25,38], our work distinguishes up to three different addressees. The difference between our classification of addressees and that of other authors is that a fourth type of addressee, identified as a “stranger” by Burén and Lunde [38] and Dolev-Cohen and Ricon [40], for example, was dismissed from our study after having been defined by our discussion groups as someone adolescents know nothing about, have never seen, have never spoken to, and have no intention of knowing or speaking to. Fifth, this scale also responds to exigencies detected in previous studies with regard to whether sexting occurs due to one’s own initiative or in response to a request [55], and with regard to whether the sender shows their own face or not in the pictures or videos [17]. Sixth, unlike many other studies, our instrument clearly defines a sexual or amorous purpose in the exchange of sexts and establishes a timeframe that minimizes potential recall bias by respondents. Seventh and last, A-SextS presents a modular structure with more relevant subscales and defining elements, as used previously to develop measures on sexting and which play a major role in empirical literature on the association between sexting, online risk-taking, and online sexual victimization [25,38]. All these elements make A-SextS a comprehensive instrument comprising most of the sexting experiences and features considered in the previous literature. A-SextS can, therefore, be regarded as a more extensive and fine-grained scale for measuring a wider variety of sexting behaviors.

Regarding A-SextS’ psychometric results, concurrent validity was supported in most comparisons by overlapping confidence intervals between prevalence estimates reported in this study and those reported in a recent meta-analytic study or in comparable individual empirical studies, considered as independent external indicators. The quasi or non-concurrency of certain comparisons with the results of some previous studies may be accounted for by two main reasons. The first is that adolescent sexting has both been found to increase over time and with an increasing mean age of participants in the sample [2,17]. The second is that the higher sexting rates found in this study may also be due to a wider set of experiences and features (e.g., degree of explicitness, other media formats) considered in the A-SextS questionnaire in comparison to previous sexting measures. Regarding the criterion validity of A-SextS, an analysis of Kendall’s Tau coefficient and various regression analyses showed that, as expected, sexting was more prevalent among older adolescents, pornography consumers, and sexually active adolescents. This consistent relationship supports the relational-sexual nature of sexting, which is commonly used by adolescents as a legitimate way to fulfil their own sexual developmental needs. Furthermore, the congruent results between the three types of regression demonstrate more robustness in the findings and improve their interpretability. For instance, the odds of sending a sext to a boyfriend/girlfriend for a sexually active adolescent is 7.6 times that for a non-sexually active adolescent (binary logistic regression); a sexually active adolescent is expected to send 4.5 more sexts a month to their boyfriend/girlfriend than a non-sexually active adolescent (Poisson regression); and a sexually active adolescent is expected to obtain a higher score of 4.1 in the SF (sending sexts to a boyfriend/girlfriend) subscale than a non-sexually active adolescent (negative binomial regression). It should be noted that the subscales were formed on the experiences and addressees of sexting. However, the modular structure of A-SextS makes it possible to use different subscales depending on the research question of the specific study. In this respect, future studies can consider different features of sexting (e.g., addressees, degree of explicitness) with the purpose of obtaining a finer-grained picture of the relationship between criterion variables and different sexting behaviors. It is important, however, to check that subscales used are essentially unidimensional [46].

The validation process of A-SextS was different from that applied in previous sexting measures [33,35] in which a factor analytic perspective was adopted to explore the latent structure of the scale. Penado et al. [35], for example, carried out a factorial analysis and grouped factors according to the medium through which sexts were communicated (i.e., mobile phone vs. social networks), with a correlation coefficient of 0.9 between factors. The issue with such a factorial structure is the difficulty in separating the functionalities of both media, which can overlap. The exchange of any sexts can be conducted via a social network and, in turn, via a mobile phone, and vice versa. In addition, both media cannot be considered latent variables that configure factorial structures supported by a prior theoretical framework. Our decision in changing the validation approach was based on two main motivations, which are interrelated. First, the lack of theoretical models in sexting literature makes it difficult to interpret the meaning and relationships between the resulting dimensions from a factor analysis. The formulation of a theoretical model is difficult to accomplish without an operationalized definition of the construct under study. A-SextS is intended to fill this gap by proposing an integrated sexting definition that covers several elements detected in previous review and empirical studies. Once the various sexting defining elements have been identified and repeatedly measured, a theory can be better constructed. Otherwise, we may run the risk of forcing the existence of certain latent structures, regardless of issues associated with extremely high factor correlations or a high number of unexplained cross-loadings. The modular structure of A-SextS clearly departs from the simple structure often sought in factor analysis studies. Each A-SextS item has been constructed as an indicator of a behavior relating to more than a single sexting element. Thus, instead of using the whole scale with a specified latent structure, either individual items or different subscales can be used according to the research purposes. For instance, the experiences and addressees of sexting were the selected elements in the criterion validity study, but other subscales may be constructed by focusing on different sexting defining elements. Currently, sexting measures appear to be closer to the Patient Reported Outcome Measures (PROMs) approach [56,57] than to measures guided by a theory-driven approach aimed at validating a known factor structure via a conventional strategy. Since sexting still cannot be considered as a construct of its own embedded in a validated theoretical system, PROMs could offer a potential solution to issues of assessment and validation regarding behavioral practices lacking in satisfactory theoretical background, as is the case of sexting.

### 4.1. Limitations and Future Research

Certain limitations to this study and future research lines should be mentioned. First, A-SextS was designed to focus on assessing active experiences of sexting, although it also considers the reception of sexts as a passive experience. Second, all participants were selected by convenience, which means that the study’s generalizability is limited. It was intended to collect data from a more heterogeneous sample in terms of age, however, the health crisis caused by Covid-19 prevented data collection at a fourth school that would have provided more sample heterogeneity. As a result, most participants were between the age of 12 and 16, with likely notable differences between them regarding the management of their own sexual intimacy, their sexual developmental needs, and their accumulated sexual experiences. Future research should apply probabilistic sampling methods to involve a wider population and to examine the characteristics of sexting by age. In addition, our discussion groups were formed by a majority of girls, which may have affected the contents of the discussion in terms of experiences and opinions on sexuality. Nonetheless, we believe this gender disparity to be inconsequential, given that the purpose of the discussion groups was limited to examining the comprehensibility and semantic validity of the scale, once the conceptualization of sexting had been properly defined. Third, the English language version of A-Sexts has not been subjected to a validation process, but is simply the product of translation by a professional linguist and native English speaker specialized in the translation of scientific and technical texts. Future lines of research should try to explore the psychometric properties of A-SextS in various different cultural contexts and at different educational stages. Fourth, A-SextS is a self-report scale, therefore answers may be influenced by social desirability and concerns about being judged, despite anonymity being clearly affirmed. Future studies should test an administration procedure in which students respond in a totally isolated scenario. Fifth, as a field research limitation and with respect to concurrent validity, we were unable to compare the prevalence estimates of some of the sexting subscales considered due to the absence of previous studies reporting the same domains. Sixth, while A-SextS conceives sexting as a practice carried out with an amorous or sexual purpose or responding to sexual objectives, our instrument does not explicitly differentiate between direct and indirect pressures, nor does it differentiate between coercive acts that may lead adolescents to take certain actions. Future research may also focus on developing instruments to assess these distinctions in the practice of sexting.

### 4.2. Implications

Despite the aforementioned limitations, our results have some important implications. The scale fills a certain gap in the field by providing researchers with a homogeneous, extensive, and objective measure of sexting that considers the riskiest characteristics of this practice (e.g., showing one’s face in pictographic sexts), and provides good evidence of content, concurrent, and criterion validity. Taking into account the abovementioned riskiest characteristics remains a priority in the assessment of sexting behavior, since the consequences of the malicious use of pictures or videos, such as non-consensual sharing of received, intimate content in which one is easily identifiable or recognizable, may be particularly harmful [58]. Furthermore, the modular structure of A-SextS will allow academics and teachers to combine or focus on the study of any experience of sexting, relationship type, multimedia content, explicitness, motivation, and inclusion of the participant’s face in content. All the features that A-SextS covers can serve as a basis for the formulation of polices and educational measures regarding how adolescents manage sexual interactions.

## 5. Conclusions

This study contributes to an enrichment of research on adolescent sexting, and validates an instrument with good psychometric properties for assessing this phenomenon. Furthermore, it aims to help break a vicious circle that, in our opinion, has characterized much of sexting research to date: a body of empirical results from poor quality sexting measures making it difficult to develop consensual theoretical explanations of the practice. Lastly, our Adolescent Sexting Scale (A-SextS) also has important implications for educational interventions since it considers different types of social relationships and their characteristics, and ethical issues such as voluntariness.

## Figures and Tables

**Figure 1 ijerph-17-08042-f001:**
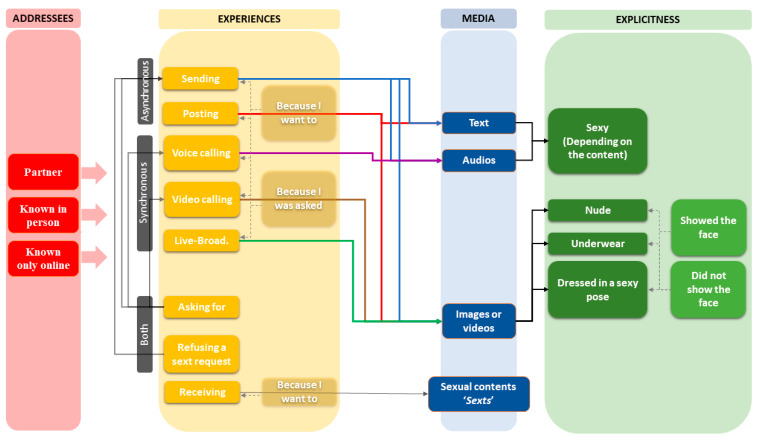
Graphical representation of the modular structure of A-SextS.

**Table 1 ijerph-17-08042-t001:** Participants’ age range distribution.

	Total Sample *n* = 579, (*n*) %	
Age (years)	(*n*)	%
11	1	0.2
12	92	16.9
13	126	23.2
14	124	22.8
15	129	23.7
16	54	9.9
17	16	2.9
18	2	0.4
Minimum	11	
Maximum	18	
Range	7	
Mean (*M*)	13.9	
Standard Deviation (*SD*)	1.3	

**Table 2 ijerph-17-08042-t002:** Proportion of category frequency responses of Adolescent Sexting Scale (A-SextS) items.

Item No.	Short-item Description	%0	%1	%2	%3	%4
1	Sent a sexy text to boyfriend/girlfriend	68.92	10.59	7.99	7.64	4.86
2	Sent a sexy text to someone known in person	68.23	16.49	7.99	4.34	2.95
3	Sent a sexy text to someone known only on the internet	85.04	9.04	3.48	1.39	1.04
4	Sent a sexy audio to boyfriend/girlfriend	87.00	5.20	3.12	2.43	2.25
5	Sent a sexy audio to someone known in person	85.76	7.12	5.38	1.39	0.35
6	Sent a sexy audio to someone known only on the internet	95.66	1.74	1.04	1.04	0.52
7	Sent naked image/video to boyfriend/girlfriend	93.75	2.26	1.74	1.74	0.52
8	Sent naked image/video to someone known in person	93.92	2.95	1.22	1.39	0.52
9	Sent naked image/video to someone known only on the internet	97.22	1.39	0.69	0.17	0.52
10	Sent underwear image/video to boyfriend/girlfriend	82.64	9.03	4.51	2.43	1.39
11	Sent underwear image/video to someone known in person	85.44	9.19	2.60	1.73	1.04
12	Sent underwear image/video to someone known only on the internet	95.32	2.77	0.69	0.52	0.69
13	Sent dressed image/video to boyfriend/girlfriend	78.78	8.87	5.91	3.30	3.13
14	Sent dressed image/video to someone known in person	79.41	12.46	4.67	1.90	1.56
15	Sent dressed image/video to someone known only on the internet	94.10	3.12	1.91	0.69	0.17
16	Sent a sexy image/video of other people to boyfriend/girlfriend	94.63	3.29	1.56	0.17	0.35
17	Sent a sexy image/video of other people to someone known in person	92.39	3.63	2.94	0.35	0.69
18	Sent a sexy image/video of other people to someone known only on the internet	95.32	2.43	1.39	0.35	0.52
19	Posted a sexy text	72.13	16.72	5.92	3.66	1.57
20	Posted naked image/video	98.26	1.04	0.17	0.35	0.17
21	Posted underwear image/video	89.06	6.60	2.43	1.56	0.35
22	Posted dressed image/video	76.47	15.40	5.02	2.42	0.69
23	Streamed naked video	98.79	0.69	0.35	0.17	0.00
24	Streamed underwear video	96.88	1.73	0.52	0.52	0.35
25	Streamed dressed video	91.51	5.89	1.56	0.52	0.52
26	Sexy voice calls with boyfriend/girlfriend	79.55	6.59	7.11	3.81	2.95
27	Sexy voice calls with someone known in person	82.35	9.00	5.19	2.42	1.04
28	Sexy voice calls with someone known only on the internet	95.32	3.29	0.69	0.35	0.35
29	Naked video call with boyfriend/girlfriend	94.29	3.11	1.38	0.87	0.35
30	Naked video call with someone known in person	96.54	1.73	1.04	0.52	0.17
31	Naked video call with someone known only on the internet	97.75	1.38	0.35	0.17	0.35
32	Underwear video call with boyfriend/girlfriend	85.27	6.24	4.68	2.60	1.21
33	Underwear video call with someone known in person	89.58	5.38	2.78	1.91	0.35
34	Underwear video call with someone known only on the internet	97.23	1.73	0.35	0.52	0.17
35	Dressed video call with boyfriend/girlfriend	82.50	6.76	4.51	3.47	2.77
36	Dressed video call with someone known in person	85.42	8.51	3.12	2.43	0.52
37	Dressed video call with someone known only on the internet	97.23	2.25	0.35	0.17	0.00
38	Asked for a sexy text to boyfriend/girlfriend	87.72	6.23	3.46	1.38	1.21
39	Asked for a sexy text to someone known in person	91.19	6.04	1.38	1.04	0.35
40	Asked for a sexy text to someone known only on the internet	96.19	2.25	1.38	0.17	0.00
41	Asked for a sexy audio to boyfriend/girlfriend	93.23	3.12	1.91	1.22	0.52
42	Asked for a sexy audio to someone known in person	94.45	3.47	1.39	0.52	0.17
43	Asked for a sexy audio to someone known only on the internet	98.79	0.52	0.35	0.17	0.17
44	Asked for naked image/video to boyfriend/girlfriend	91.36	4.49	1.90	1.73	0.52
45	Asked for naked image/video to someone known in person	93.09	3.45	1.55	1.21	0.69
46	Asked for naked image/video to someone known only on the internet	95.50	2.25	1.21	0.87	0.17
47	Asked for underwear image/video to boyfriend/girlfriend	88.41	6.06	2.77	2.42	0.35
48	Asked for underwear image/video to someone known in person	92.40	3.97	1.90	1.38	0.35
49	Asked for underwear image/video to someone known only on the internet	96.37	1.90	0.69	0.87	0.17
50	Asked for dressed image/video to boyfriend/girlfriend	86.98	6.77	2.60	2.78	0.87
51	Asked for dressed image/video to someone known in person	91.71	5.18	1.73	0.86	0.52
52	Asked for dressed image/video to someone known only on the internet	96.89	1.38	1.04	0.35	0.35
53	Asked for a sexy voice call to boyfriend/girlfriend	91.70	3.29	1.90	2.08	1.04
54	Asked for a sexy voice call to someone known in person	92.06	4.84	1.38	1.38	0.35
55	Asked for a sexy voice call to someone known only on the internet	97.41	1.21	0.86	0.35	0.17
56	Asked for a sexy video call to boyfriend/girlfriend	90.83	3.29	2.08	2.42	1.38
57	Asked for a sexy video call to someone known in person	93.26	2.94	1.90	1.55	0.35
58	Asked for a sexy video call to someone known only on the internet	97.92	0.35	0.69	0.87	0.17
59	Refused to send sexy contents to boyfriend/girlfriend	88.75	7.56	2.11	1.05	0.53
60	Refused to send sexy contents to someone known in person	83.95	10.05	3.53	1.76	0.71
61	Refused to send sexy contents to someone known only on the internet	84.01	8.44	3.69	2.11	1.76
62	Received a sexy content from boyfriend/girlfriend	74.25	13.05	6.00	3.70	3.00
63	Received a sexy content from someone known in person	71.48	16.55	6.87	3.35	1.76
64	Received a sexy content from someone known only on the internet	80.11	11.97	3.87	2.64	1.41

Note: The categories, from 0 to 4, correspond to “never”, “between 1 and 3 times a month”, “between 1 and 3 times a week”, “every or almost every day”, and “several times a day”, respectively.

**Table 3 ijerph-17-08042-t003:** Essential unidimensionality of the subscales.

Subscale	Nº Items	PA	Kaiser	λ_1_ / λ_2_	h^2^	Alpha	Omega
SF	10	1	1	7.2	0.63	0.88	0.94
SK	10	1	1	7.1	0.61	0.85	0.94
SI	10	1	1	9.2	0.72	0.87	0.96
PS	6	1	1	4.5	0.55	0.62	0.88
AF	7	1	1	8.6	0.73	0.89	0.95
AK	7	1	1	9.7	0.74	0.88	0.95
AI	7	1	1	10.1	0.75	0.85	0.95
R	3	1	1	3.9	0.59	0.73	0.81
RS	3	1	1	5.5	0.72	0.76	0.88

Note: SF = sending sexts to a boyfriend/girlfriend; SK = sending sexts to someone known in person; SI = sending sexts to someone known only on the internet; PS = posting or live-streaming pictographic content; AF = asking for sexts from a boyfriend/girlfriend; AK = asking for sexts from someone known in person; AI = asking for sexts from someone known only on the internet; R = receiving sexts; RS = refusing to send a requested sext; PA = parallel analysis; Kaiser = eigenvalue-higher-than-one rule; λ_1_ / λ_2_ = ratio between the first and second eigenvalues; h^2^ = average communality; Alpha = Cronbach’s Alpha; Omega = McDonald’s Omega.

**Table 4 ijerph-17-08042-t004:** Kendall’s Tau coefficient.

Subscale	Age	Pornography Consumption	Sexual Intercourse
SF	0.148 ***	0.160 ***	0.412 ***
SK	0.153 ***	0.190 ***	0.262 ***
SI	0.064	0.198 ***	0.197 ***
PS	0.121 ***	0.056	0.314 ***
AF	0.099 *	0.190 ***	0.398 ***
AK	0.066	0.183 ***	0.270 ***
AI	0.095 *	0.146 ***	0.151 ***
R	0.207 ***	0.190 ***	0.375 ***
RS	0.135 ***	−0.002	0.241 ***

Note: * *p* < 0.05; *** *p* < 0.001. SF = sending sexts to a boyfriend/girlfriend; SK = sending sexts to someone known in person; SI = sending sexts to someone known only on the internet; PS = posting or live-streaming pictographic content; AF = asking for sexts from a boyfriend/girlfriend; AK = asking for sexts from someone known in person; AI = asking for sexts from someone known only on the internet; R = receiving sexts; RS = refusing to send a requested sext.

**Table 5 ijerph-17-08042-t005:** Regression analyses.

	Binary Logistic Regression		Poisson Regression		Negative Binomial Regression	
Subscale	Pornography Consumption	Sexual Intercourse	Pornography Consumption	Sexual Intercourse	Pornography Consumption	Sexual Intercourse
SF	2.622 *** (1.610, 4.331)	7.595 *** (4.062, 15.113)	2.021 *** (1.940, 2.105)	4.520 *** (4.348, 4.698)	1.759 * (1.181, 2.626)	4.088 *** (2.663, 6.422)
SK	2.591 *** (1.692, 4.013)	3.163 *** (1.896, 5.393)	3.429 *** (3.251, 3.617)	4.951 *** (4.715, 5.199)	2.125 *** (1.469, 3.089)	3.199 *** (2.092, 5.062)
SI	3.522 *** (2.069, 6.114)	2.667 *** (1.527, 4.627)	6.574 *** (5.954, 7.268)	2.800 *** (2.580, 3.038)	3.401 *** (1.817, 6.474)	2.797 ** (1.386, 6.224)
PS	1.909 ** (1.211, 3.042)	4.086 *** (2.497, 6.749)	2.196 *** (1.991, 2.422)	4.271 *** (3.886, 4.694)	1.775 * (1.171, 2.713)	3.205 *** (2.068, 5.067)
AF	3.431 *** (1.987, 6.066)	7.641 *** (4.351, 13.769)	4.372 *** (4.075, 4.691)	9.159 *** (8.576, 9.784)	2.893 *** (1.682, 5.002)	9.315 *** (4.940, 18.327)
AK	3.171 *** (1.872, 5.470)	4.568 *** (2.668, 7.866)	8.285 *** (7.554, 9.096)	13.572 *** (12.482, 14.766)	3.695 *** (2.071, 6.693)	7.041 *** (3.626, 14.824)
AI	4.103 *** (1.891, 9.308)	2.625 ** (1.260, 5.358)	12.497 *** (10.523, 14.772)	2.626 *** (2.320, 2.971)	7.339 *** (2.627, 23.074)	3.735 * (1.298, 13.674)
R	2.737 *** (1.768, 4.298)	5.215 *** (2.976, 9.577)	3.946 *** (3.679, 4.234)	5.618 *** (5.260, 6.000)	2.537 *** (1.857, 3.486)	3.450 *** (2.500, 4.815)
RS	1.864 * (1.130, 3.115)	2.800 *** (1.667, 4.703)	1.548 *** (1.404, 1.706)	3.450 *** (3.139, 3.791)	1.728 * (1.035, 2.943)	2.885 *** (1.710, 5.000)

Note: * *p* < 0.05; ** *p* < 0.01; *** *p* < 0.001. SF = sending sexts to a boyfriend/girlfriend; SK = sending sexts to someone known in person; SI = sending sexts to someone known only on the internet; PS = posting or live-streaming pictographic content; AF = asking for sexts from a boyfriend/girlfriend; AK = asking for sexts from someone known in person; AI = asking for sexts to someone known only on the internet; R = receiving sexts; RS = refusing to send a requested sext. The regression coefficients are shown exponentiated (ExpB). 95% confidence intervals are shown in brackets.

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
