# Peer review of "Development and Validation of the Adolescent Sexting Scale (A-SextS) with a Spanish Sample"

_ijerph, 2020, doi:10.3390/ijerph17218042_

Round 1

Reviewer 1 Report

Thank you for giving me the opportunity to review this paper. This paper has the aim to fill a large gap in the current literature regarding sexting. The questionnaire is very detailed and allows us to study sexting as a whole. However I still wonder about the implementation of such a questionnaire, particularly in schools, in view of its length but it seems that the authors consider that the use of subscale is possible (maybe you should highlight it before in the text). I am aware that there are a lot of elements to consider and the article is already dense, but I find that there is a lot of information that is left in supplementary documents which requires for the readers to find certain information that, in my opinion, are important to understand the process. Perhaps you could further explain some points in the article itself. Some of my questions were answered with the supplemental documents but I would have preferred to read them directly in the manuscript.

L34: what does nature of sexting mean?

When you present the new research approach with the normativity of sexting, you should link this to the new approach of prevention and present what normativity mean (to avoid shortcuts such as normative = the norm for all young people (see Döring https://cyberpsychology.eu/article/view/4303/3352).

L46 : You talk about the risks of sexting. Be careful, in my opinion, the acts listed such as sextortion, grooming, etc. are related but this is not a direct risk of what is defined as sexting. The risks are dissemination, pressure, harassment (including cyber) that follows. Sextortion is very special and will be treated differently, particularly legally. For sextortion, we have companies that are behind the pressure / harassment and for grooming, pedophiles are often involved. If links can be made between sexting and other online sexual experiences, I would not say sexting leads to that, even if personal sexual images are involved.

L46 : non consensual distribution of sexts. I understand but specify that it is the image, for example, of another person that is distributed.

L71 : you talked about voluntary sexting. Do you include the indirect pressure that may be experienced? There are several studies that have looked at this pressure, especially among girls (e.g. Lippman & Campbell, 2014).

L87 : I do not understand the “however” in the final sentence of this paragraph. The whole paragraph above talks about consensus issues in terms of measurement... What do you mean?

Did the EDIMA talk about images of oneself? It would be interesting to talk about the very first study on sexting in 2008 (sex tech survey) because it would seem that this questionnaire is based on the same categories used in 2008.

L102, but also in your aims, you say that the other technologies were not covered by the studies, but be careful, the vast majority of exchanges are done with cellphone on social networking applications. I agree to that it is very interesting to know everything, but in my opinion, it is not the main force for improving our knowledge.

L106 : Be careful, I think there is a whole continuum between voluntary and coerced sexting, especially indirect pressure (Do you love me?)? And in coerced sexting do you include unwanted receiving, in other words the fact of receiving without asking for example?

L112 the point a) is not clear for me.

Based on your questions, even if it was on nonconsensual sexting, you could refer to a study which was conducted among young adults because it has a lot in common with your questions: the addressee, sexual content, motivations, face, etc. https://pubmed.ncbi.nlm.nih.gov/32594970/

L130 : In my opinion, between 11-18 years old there are strong differences in terms of sexting and sexuality ...

L137: how many schools have refused? How long did the questionnaire last?

L139 : were the pupils able to refuse to participate? If so, how many did?

L155 : it is not common to cite an article under review... is it at least in press? Readers cannot refer to it, you take the risk that this article will appear first, how do we know which articles are considered in your review?

L161 : What does the goal or purpose refer to? Article? Sexting?

L172 : you had 14 out of 21 girls, we know that sexting and sexuality are experienced differently according to the gender, is this a bias?

L185 : how did you concretely manage to collect the opinions, difficulties / misunderstandings of young people with the 67 questions? Were there any questions on this in addition to the survey?

L225 The second quotes are missing to close the sentence

L230 What is the difference between streaming and video call, both are live

L233 : before you said that the audio was not common but you seem to have kept it anyway, what did you use this information for? To reduce the number of questions? I think information is missing on what you had to change after the group discussion and pilot study. I did not quite understand how the pilot study was able to refine the questions.

I do not understand the “two protagonists (oneself and another)”, since in your definition you indicate “with another person”.

L240: It may be that this was requested and the person responded positively without constraint, I think that this distinction does not differentiate clearly between coercion, pressure, including indirect pressure and consent. It is a limitation.

L244 : Why is there a refined version? What has been changed? Is it following the pilot study?

Table 1 : it is disturbing to see only text or audio without sexy / or without a reference to sexual content while sexy is indicated in your questionnaire.

I do not quite understand the difference between sending and posting, posting on a social network? In public?

Figure 1 : Did you distinguish between wanted and unwanted reception? In your figure we only see “Because I want to”, so it is only wanted reception, is that correct?

L257: did you have a safety net? I think you have students under the age of 14 who are going to answer questions about their sexuality and about non-consensual sexual intercourse. How did you follow up?

L315 : by purpose do you mean the purpose of the sending? The motivations ?

L308 : Based on the presentation of your methods, and the aim of validating a questionnaire, I do not understand the part 3.1.1 in your results. What are you trying to say? I think this part is more a support to your aim. Maybe it should appear before in the text

Likewise for point 3.1.2, you refer us to supplementary document S7, but I think that you either have to tell us what these discussion groups concretely made you modify or leave it in the supplement totally.

L336 : In terms of the age of the respondents, are you similar to these studies? The differences could be even greater given your average age. How to explain it?

L526 : I think you should talk about the age of your respondents in your limitations. Even though you have an average age of 13.4 years, you still have difference in terms of sexual experiences between, for example, an 11-year-old and a 15-year-old.

References: 25 and 39 are the same.

Author Response

Comments and Suggestions for Authors

Comment 1.

Thank you for giving me the opportunity to review this paper. This paper has the aim to fill a large gap in the current literature regarding sexting. The questionnaire is very detailed and allows us to study sexting as a whole. However I still wonder about the implementation of such a questionnaire, particularly in schools, in view of its length but it seems that the authors consider that the use of subscale is possible (maybe you should highlight it before in the text). I am aware that there are a lot of elements to consider and the article is already dense, but I find that there is a lot of information that is left in supplementary documents which requires for the readers to find certain information that, in my opinion, are important to understand the process. Perhaps you could further explain some points in the article itself. Some of my questions were answered with the supplemental documents but I would have preferred to read them directly in the manuscript.

Authors' response: The authors appreciate the time and effort you have dedicated to providing valuable suggestions on our manuscript. All of them have been answered in a reasoned manner and the manuscript has been revised accordingly. We believe that the contents and the clarity of our paper are much improved in the revised version.

Comment 2.

L34: what does nature of sexting mean?

Authors' response: In accordance with your comment and since the term "nature" is too ambiguous, we have rewritten this sentence as “However, the paucity of theoretical explanations of consensus on this phenomenon has left an open debate about the motivations, opportunities and risks of this practice” (L 35).

Comment 3.

When you present the new research approach with the normativity of sexting, you should link this to the new approach of prevention and present what normativity mean (to avoid shortcuts such as normative = the norm for all young people (see Döring https://cyberpsychology.eu/article/view/4303/3352).

Authors' response: Thank you for this suggestion. As requested, we have included an explanation of the normative discourse, linking it to the approach of prevention and risk reduction. Specifically, we have written this suggestion as “From this perspective, sexting is understood as just another form of sexual expression in the context of contemporary sexual or romantic relations, which can, in fact, be carried out 'safely' by young people when appropriate strategies are applied to reduce possible negative consequences [10]” (see L 38). The reference you have provided has also been included. Thank you.

Comments 4 and 5

L46 : You talk about the risks of sexting. Be careful, in my opinion, the acts listed such as sextortion, grooming, etc. are related but this is not a direct risk of what is defined as sexting. The risks are dissemination, pressure, harassment (including cyber) that follows. Sextortion is very special and will be treated differently, particularly legally. For sextortion, we have companies that are behind the pressure / harassment and for grooming, pedophiles are often involved. If links can be made between sexting and other online sexual experiences, I would not say sexting leads to that, even if personal sexual images are involved.

L46 : non-consensual distribution of sexts. I understand but specify that it is the image, for example, of another person that is distributed.

Authors' response: We are very grateful for your reflections on this matter. As requested, we have rewritten this sentence as “A common risk is the intentional, non-consensual distribution of third-party sexual images, whose prevalence among youths has been shown to lie between 8.4 and 15.6% [2]” (L 51). We hope that, with the current version of the sentence, this matter has been resolved.

Comment 6.

L71 : you talked about voluntary sexting. Do you include the indirect pressure that may be experienced? There are several studies that have looked at this pressure, especially among girls (e.g. Lippman & Campbell, 2014).

Authors' response: Thank you for your suggestion and thank you for having sent us this reference. As suggested, we have completed the sentence as “Although sexting is often thought of as a voluntary practice, most studies do not specify it as such, nor consider the indirect pressure to exchange sexts that adolescents may feel or receive” (L 76). The reference you have provided has also been included.

Comment 7.

L87: I do not understand the “however” in the final sentence of this paragraph. The whole paragraph above talks about consensus issues in terms of measurement... What do you mean?

Authors' response: We agree with your suggestion. Accordingly, we have eliminated 'however' as an adverb with an adverse sense and have rewritten the phrase as “Lastly, the most notable methodological limitation of research on sexting is the absence of a consensus on its measurement, especially in adolescents” (L 93).

Comment 8.

Did the EDIMA talk about images of oneself? It would be interesting to talk about the very first study on sexting in 2008 (sex tech survey) because it would seem that this questionnaire is based on the same categories used in 2008.

Authors' response: In response to the first question, the EDIMA-scale does ask about the sending and posting of one’s own images/videos. In response to your suggestion, however, we would like to clarify that the questionnaire applied in the Sex and Tech study did not follow a validation process. This is why in the section "1.1 Existing Sexting Measures" we do not refer to the Sex and Tech Survey (2008), since we only present the sexting measures that have undergone a validation process. Therefore, we have considered it more appropriate to rewrite the subsection title as “1.1. Existing Validated Sexting Measures” (L 96).

Comment 9.

L102, but also in your aims, you say that the other technologies were not covered by the studies, but be careful, the vast majority of exchanges are done with cell phone on social networking applications. I agree to that it is very interesting to know everything, but in my opinion, it is not the main force for improving our knowledge.

Authors' response: We agree with this reflection. This phrase and the following serve to remind us that some of the conceptual issues mentioned throughout the “Introduction” section have not yet been considered or have not found consensus in the design of existing validated sexting instruments. Regarding the method of transmission of sexts, the scales we present (ESC, SBS, EDIMA) do not cover, for example, live streaming platforms (e.g. Vimeo, Tumblr or Twitch) or chats internal to certain video games. We use this simply as an example of conceptual gap, because we think that any measure of sexting should holistically consider the method of transmission and should not be restricted only to mobile phones or social network sites. In any case, we appreciate you sharing this reflection with us.

Comment 10

L106: Be careful, I think there is a whole continuum between voluntary and coerced sexting, especially indirect pressure (Do you love me?)? And in coerced sexting do you include unwanted receiving, in other words the fact of receiving without asking for example?

Authors' response: We acknowledge your suggestion. To address this issue, we have rewritten the sentence as "Lastly, voluntariness has not been expressly considered, making it impossible to distinguish between fine-grained degrees of voluntariness in sexting, such as intentional sexting, unwanted but consensual sexting, and coerced sexting" (L 121). We consider these reflections extremely relevant.

Comment 11.

L112: the point a) is not clear for me.

Authors' response: We have tried to clarify this point by reformulating the sentence as “a) focusing on active sexting, covering a wide range of online behaviours, some of which not considered to date (e.g. posting, streaming)” (L 129).

Comment 12.

Based on your questions, even if it was on nonconsensual sexting, you could refer to a study which was conducted among young adults because it has a lot in common with your questions: the addressee, sexual content, motivations, face, etc. https://pubmed.ncbi.nlm.nih.gov/32594970/

Authors' response: We sincerely thank you for this reference. The reference you have provided to us reinforces arguments about the non-consensual distribution of received sexts and the importance of taking into consideration aspects such as the addresses of sects or showing one’s face in pictographic sexts. This is considered in the “Implications” section (L 644) as "Taking into account the abovementioned riskiest characteristics remains a priority in the assessment of sexting behaviour, since the consequences of the malicious use of pictures or videos, such as non-consensual sharing of received, intimate content in which one is easily identifiable or recognizable, may be particularly harmful [17]”.

Comments 13 and 32

L130: In my opinion, between 11-18 years old there are strong differences in terms of sexting and sexuality.

L526 : I think you should talk about the age of your respondents in your limitations. Even though you have an average age of 13.4 years, you still have difference in terms of sexual experiences between, for example, an 11-year-old and a 15-year-old.

Authors' response: Since the age range, a priori, is certainly wide, we have decided to include a table (Table 1, L 152) with the distribution of the ages of the adolescents who make up the sample (as also indicated by Reviewer 2). This table shows that most of the sample are between 12 and 16 years of age (96.5%). Likewise, as requested, we have incorporated this remark in the “Limitations” section as “Second, all participants were selected by convenience, which means that the study’s generalizability is limited. It was intended to collect data from a more heterogeneous sample in terms of age, however, the health crisis caused by Covid-19 prevented data collection at a fourth school that would have provided more sample heterogeneity. As a result, most participants were between the age of 12 and 16, with likely notable differences between them regarding the management of their own sexual intimacy, their sexual developmental needs, and their accumulated sexual experiences. Future research should apply probabilistic sampling methods to involve a wider population and to examine the characteristics of sexting by age” (L 613). We appreciate your observation once again.

Comment 14.

L137: how many schools have refused? How long did the questionnaire last?

Authors' response: No school refused to participate in the study. For this study, we contacted schools that had showed an interest in collaborating in one of our previous research studies on sexting, but were unable to participate at the time. Therefore, for this occasion, we directly contacted the principals of these schools and all of them agreed to collaborate in this research. It should be noted that the school where the pilot test was carried out decided to collaborate only at this stage and not in the final collection of questionnaires due to organizational issues specific to the school. In accordance with your comment, this information has been included in the “Procedure” section as "One of the schools decided to collaborate only in carrying out the pilot test of the scale, whereas the other two schools participated in the final data collection” (L 181).

We have also expanded the following sentence to indicate how long it took teenagers to complete the questionnaire: “The questionnaire was administered to the participating adolescents in their usual classrooms, during regular class hours, and took approximately 40 minutes” (L 186).

Comment 15

L139 : were the pupils able to refuse to participate? If so, how many did?

Authors' response: Yes, adolescents were informed that participation in this research project was entirely voluntary, and no negative consequences would result from them abandoning or not participating in it. This information was provided to both parents, via the letter of consent (provided to the associate editor), and students via the instructions given. To address your suggestion more properly, we have expanded the “Procedure” section with the sentence "The adolescents were informed that participation in this research project was entirely voluntary, and no negative consequences would result from them abandoning or not participating in it. Ultimately, no adolescent abandoned or refused to participate in the project" (L 191).

Comment 16.

L155: it is not common to cite an article under review... is it at least in press? Readers cannot refer to it, you take the risk that this article will appear first, how do we know which articles are considered in your review?

Authors' response: Thank you for pointing this out. We agree with your comment. To avoid this scenario, we have created and cited (L 207) an appendix that includes the 79 revised studies.

Comment 17 and 29

L161 : What does the goal or purpose refer to? Article? Sexting?

L315 : by purpose do you mean the purpose of the sending? The motivations?

Authors' response: Thank you for pointing this out. As regards your first questions, point (d) “whether a goal or purpose was specified” refers to the sexting measure, namely, whether the primary items specified the purpose of sexting. Consequently, we have rewritten this sentence as “d) whether a goal or purpose for sexting was specified” (L 213).

Authors' response: Likewise, regarding your second question, we have also modified the sentence to be “and only 2 studies defined the purpose of sexting among primary items" (L 402). We hope that in the current version of the manuscript, this matter has been resolved. Also in response to Comment 17.

Comment 18.

L172 : you had 14 out of 21 girls, we know that sexting and sexuality are experienced differently according to the gender, is this a bias?

Authors' response: We have incorporated this aspect in the “Limitations” section by remarking, "In addition, our discussion groups were formed by a majority of girls which may have affected the contents of the discussion in terms of experiences and opinions on sexuality. Nonetheless, we believe this gender disparity to be inconsequential, given that the purpose of the discussion groups was limited to examining the comprehensibility and semantic validity of the scale, once the measurement construct had been properly defined". (L 620). Certainly, we believe that this gender disparity would have been consequential if the discussion groups had aimed to define sexting itself.

Comments 19, 22 and 31

L185 : how did you concretely manage to collect the opinions, difficulties / misunderstandings of young people with the 67 questions? Were there any questions on this in addition to the survey?

L233 : before you said that the audio was not common but you seem to have kept it anyway, what did you use this information for? To reduce the number of questions? I think information is missing on what you had to change after the group discussion and pilot study. I did not quite understand how the pilot study was able to refine the questions.

Likewise for point 3.1.2, you refer us to supplementary document S7, but I think that you either have to tell us what these discussion groups concretely made you modify or leave it in the supplement totally.

Authors' response: We are very grateful for your suggestions on this matter. This aspect had not been sufficiently emphasized in our article. As a clarification, what the discussion groups considered not common was the posting and retransmission of sexy audios, but not sending sexy audios to a specific person. Sending sexy audios was the most common experience reported by adolescents regarding the audio format.

In order to provide a more specific response to your three comments, we have now expanded this paragraph (L 271) and the subsequent paragraph with "[Explanation about the organization of the discussion groups]. The process resulted in several changes in wording of certain items and the deletion of others that were not deemed relevant or were considered uncommon. For example, the adjective used in items specifying the explicitness of the media content was changed to improve comprehension of the items by adolescents (e.g. 'sending audios of a sexual nature' was changed to 'sending sexy audios'). The expression initially used to refer to sexts featuring someone else was also changed (e.g. 'I have sent an image or video where other nudes appear' was changed to 'I have sent a sexy image or video featuring someone else'). Certain terms were also adapted to incorporate adolescent jargon (e.g. 'I have broadcast a video' was changed to 'I have live-streamed video'). Items that were not deemed relevant or were considered uncommon were deleted (e.g. posting or live-streaming sexy audios, asking someone to do live broadcasts nude, in underwear or dressed and in a sexy pose).

Finally, A-SextS’ updated list of 67 questionnaire items was pilot-tested on 96 secondary school pupils. After completing the pilot-questionnaire, the adolescents were asked about the readability and comprehension of the questionnaire, initiating a brief oral discussion between the researcher and the adolescents. Written notes were taken for decision-making purposes. The pilot test provided useful insights as to how to improve the instructions, appearance and format of the questionnaire. For example, the sociodemographic questions section was moved to the end of the questionnaire to avoid a possible fatigue effect at the time of filling it in and to prevent its answers from being conditioned by having provided such information previously. The items in the questionnaire were changed from a bulleted or numbered format (e.g. 'I have sent a sexy text message to: (a) my boy/girlfriend, and so on, followed by the frequency scale for each one) to an unnumbered format with a full text sentence. A reminder of the basic instructions in the top margin of the scale was also added to the final version. Ambiguous items were also discussed with the pupils and modified where deemed necessary. These pilot test participants were not included in the final sample. The final version of A-SextS was composed of 64 items." (L 233 to 257)

Comment 20.

L225 The second quotes are missing to close the sentence

Authors' response: Corrected (L 299). Thank you.

Comment 21.

L230 What is the difference between streaming and video call, both are live.

Authors' response: In the A-SextS Scale, a video call refers to simultaneous, two-way communication with a single receiver (e.g. with your boy/girlfriend), while streaming experiences refer to a simultaneous, two-way scenario but with a wider audience. In the case of streaming, it is common for a single person (who has initiated the streaming) to be the one exposed, while the "audience" is generally not displayed, sometimes remains anonymous, and may often communicate exclusively via a written chat function. Therefore, not everyone involved is exposed in the same way. Spanish teenagers often call this "doing a direct" (hacer un directo), meaning “doing a live (broadcast)”, and usually do this through platforms or social networks such as Vimeo, Periscope, Tumblr, Instagram or Twitch. For Spanish teenagers, the distinction between “streaming”, or “live broadcasting”, and “video calling” is clear. We hope we have clarified your doubt.

Comment 23.

I do not understand the “two protagonists (oneself or another)”, since in your definition you indicate “with another person”.

Authors' response. The term "protagonist" refers to the person represented or featuring in the sexts. A-SextS distinguishes between the exchange of content in which the adolescent addressed by the questionnaire appears from the exchange of content in which that adolescent does not appear. Since this distinction could lead to some confusion, we have decided to modify the parenthesis as "(oneself or another person)" (L 305).

Comment 24.

L240: It may be that this was requested and the person responded positively without constraint, I think that this distinction does not differentiate clearly between coercion, pressure, including indirect pressure and consent. It is a limitation.

Authors' response: We have included in the “Limitations” section that our instrument does not explicitly differentiate between direct and indirect pressures, nor does it differentiate between coercive acts that may lead adolescents to take certain actions. Finally, the importance of developing instruments to assess these sorts of sexting issues has been emphasized as a future research line. Specifically, we have included the sentence as “Sixth, while A-SextS concepts sexting as a practice carried out with an amorous or sexual purpose or responding to sexual objectives, our instrument does not explicitly differentiate the existence of direct or indirect pressures, nor does it differentiate coercive practices that may lead adolescents to take certain actions. Future research may also focus on developing instruments to assess these distinctions in the practice of sexting” (L 634)

Comment 25.

L244 : Why is there a refined version? What has been changed? Is it following the pilot study?

Authors' response: Only some of the socio-demographic questions (last page) were changed in the refined version of the questionnaire. In particular, we present a more appropriate wording of questions 4, 8, 9 and 10, since questions with closed-answers should consider all possible answer choices. We have added a note in Appendix 4 clarifying this issue. The added note has been written as “Note: We present a more appropriate wording of questions 4, 8, 9 and 10 of the socio-demographic section, since questions with closed-answers should consider all possible answer choices” (L 693).

Comment 26.

Table 1 : it is disturbing to see only text or audio without sexy / or without a reference to sexual content while sexy is indicated in your questionnaire.

Authors' response: Thanks for pointing this out. Errors were included in Table 1 by incorporating short descriptions of the items. We have modified all items with this issue (1 to 6, 16 to 19, 26 to 28, 38 to 43, 53 to 64).

Comment 27.

I do not quite understand the difference between sending and posting, posting on a social network? In public?

Authors' response: Yes, the A-SextS asks about sending sexual content to one specific person (e.g. boyfriend or girlfriend), while posting refers to making a sext public or making it available to a wider audience (e.g. a network of friends). The A-SextS specifies the means through which this action is carried out (the internet) but does not specify through which exact medium, since this might limit responses by considering as possible spaces only social network sites and not platforms, videogame chats, smart watches, and many other possibilities. This is represented in supplementary material S5. We hope we have clarified your doubt.

Comment 27.

Figure 1 : Did you distinguish between wanted and unwanted reception? In your figure we only see “Because I want to”, so it is only wanted reception, is that correct?

Authors' response: Yes, participants may or may not mark the emoticon indicating that the reception of the sexts was desired, since for adolescents it was not usual to ask whether the other person wanted to receive any particular type of content (see the discussion group results). If the box is not circled it means that reception was not desired, expected or wanted. We hope we have clarified your doubt.

Comment 28.

L257: did you have a safety net? I think you have students under the age of 14 who are going to answer questions about their sexuality and about non-consensual sexual intercourse. How did you follow up?

Authors' response: The ethical procedure and the information on the content of the questionnaire were provided to parents, via the letter of consent, administrators of the school and tutors in meetings. No agent required clarification or suggested modifying any of the questions. These aspects are addressed in lines 179 to 196.

Comment 30.

L308 : Based on the presentation of your methods, and the aim of validating a questionnaire, I do not understand the part 3.1.1 in your results. What are you trying to say? I think this part is more a support to your aim. Maybe it should appear before in the text

Authors' response: Subsection 3.1.1. gives the results of the extensive review of sexting measures that allowed this study’s authors to formulate and offer constructive criticism, to identify a wide range of conceptual reference elements to constitute our own operational definition of sexting and to create an initial pool of domains and items for use in the discussion groups. The results and criticism of these measures are presented in the Introduction section. The first paragraph of subsection 2.3.1. explains the purpose of the literature review and the expected utility of its results (L 207). We hope we have clarified this doubt.

Comment 31.

L336 : In terms of the age of the respondents, are you similar to these studies? The differences could be even greater given your average age. How to explain it?

Authors response: In the "Discussion" section, we offer plausible explanations of the quasi or non-concurrency of our prevalences with those of some previous studies. As you mentioned, in the first plausible explanation, we have highlighted that these prevalence differences may be due to the mean age of participants in the sample, as different meta-analysis have demonstrated. This is represented in L 557.

Comment 33.

References: 25 and 39 are the same.

Authors response. Thank you. Corrected.

Reviewer 2 Report

The present work contributes much interest to the study of the behavior of adolescents in relation to the practice of sexting. It is a very necessary topic to understand the nature of sexting and the experiences that adolescents have in exchanging messages of sexual content. In addition, although there are several instruments for evaluating sexting; few are specific to be applied with adolescents, so this manuscript has the value of presenting a new instrument that has an integrative conceptual approach that starts from the review of an important number of previous studies, and is also current and specific to the population Teen.

In addition, this work performs a study with an acceptable sample (N = 579). It is a study carried out in a region of Spain, although the authors should indicate the reasons why they included students between the ages of 11 and 12 among the participants. Likewise, it would be convenient to include descriptive statistics that show frequency statistics for each year of age of the participants (11 to 18 years).

On the other hand, the document is successful in the use of statistical tests and in the research methodology used. Its results are acceptable and indicate that it is an instrument that has notable strengths in terms of its content and face validity. It also provides a modular structure with its own subscales and definition elements that collect a wide variety of behaviors related to sexting. It also presents an acceptable concurrent validity and an adequate sustained criterion validity in the statistical results of the Kendall's Tau coefficient and the regression analyzes.

In addition, it presents a fairly complete review of the state of the art with a large number of current works on the subject.

Author Response

Comments and Suggestions for Authors

Comment 1. The present work contributes much interest to the study of the behaviour of adolescents in relation to the practice of sexting. It is a very necessary topic to understand the nature of sexting and the experiences that adolescents have in exchanging messages of sexual content. In addition, although there are several instruments for evaluating sexting; few are specific to be applied with adolescents, so this manuscript has the value of presenting a new instrument that has an integrative conceptual approach that starts from the review of an important number of previous studies, and is also current and specific to the population Teen.

Authors response. We are grateful for the valuable suggestions provided. All of them have been considered and the manuscript has been revised accordingly.

Comment 2. In addition, this work performs a study with an acceptable sample (N = 579). It is a study carried out in a region of Spain, although the authors should indicate the reasons why they included students between the ages of 11 and 12 among the participants. Likewise, it would be convenient to include descriptive statistics that show frequency statistics for each year of age of the participants (11 to 18 years).

Authors response. Thank you for pointing this out. We decided to include adolescents aged 11-12 and 17-18 for the purpose of working with as heterogeneous a sample as possible, and also to consider the territorial location of the schools, i.e. in metropolitan or rural areas. However, the health crisis caused by Covid-19 prevented data collection at a fourth school that would have provided even more sample heterogeneity. This has been addressed in both the “Method” (L 148) and “Limitations” sections (L 614 to 619).

Additionally, as requested, we have included a table with descriptive statistics showing the frequency distribution of the participants' ages, as well as minimum, maximum, range, mean, and standard deviation values. This table indicates that most of the sample are between 12 and 16 years of age (96.5%). Likewise, we have incorporated this remark in the “Limitations” section as “Second, all participants were selected by convenience, which means that the study’s generalizability is limited. It was intended to collect data from a more heterogeneous sample in terms of age, however, the health crisis caused by Covid-19 prevented data collection at a fourth school that would have provided more sample heterogeneity. As a result, most participants were between the age of 12 and 16, with likely notable differences between them regarding the management of their own sexual intimacy, their sexual developmental needs, and their accumulated sexual experiences”.

Comment 3. On the other hand, the document is successful in the use of statistical tests and in the research methodology used. Its results are acceptable and indicate that it is an instrument that has notable strengths in terms of its content and face validity. It also provides a modular structure with its own subscales and definition elements that collect a wide variety of behaviours related to sexting. It also presents an acceptable concurrent validity and an adequate sustained criterion validity in the statistical results of the Kendall's Tau coefficient and the regression analyzes.

In addition, it presents a fairly complete review of the state of the art with a large number of current works on the subject.

Authors response. We greatly appreciate your comments.

Round 2

Reviewer 1 Report

I am impressed with the revision and your responses to my questions / comments. I find your paper much clearer. Congratulations.